# Close Temporal Relationship between Oscillating Cytosolic K^+^ and Growth in Root Hairs of *Arabidopsis*

**DOI:** 10.3390/ijms21176184

**Published:** 2020-08-27

**Authors:** Xiangzhong Sun, Yuping Qiu, Yang Peng, Juewei Ning, Guangjie Song, Yanzhu Yang, Mengyu Deng, Yongfan Men, Xingzhong Zhao, Yichuan Wang, Hongwei Guo, Yanqing Tian

**Affiliations:** 1Department of Materials Science and Engineering, Southern University of Science and Technology, Shenzhen 518055, China; sunxz@sustech.edu.cn (X.S.); nicky19921004@163.com (J.N.); sgjsgj912@163.com (G.S.); 11710337@mail.sustech.edu.cn (Y.Y.); dengmy@sustech.edu.cn (M.D.); 2School of Physics and Technology, Wuhan University, Wuhan 430072, China; xzzhao@whu.edu.cn; 3Institute of Plant and Food Science and Department of Biology, Southern University of Science and Technology, Shenzhen 518055, China; ypingqiu@163.com (Y.Q.); eric840278845@163.com (Y.P.); wangyc@sustech.edu.cn (Y.W.); 4Harbin Institute of Technology, Harbin 150001, China; 5Department of Biology, Faculty of Science, Hong Kong Baptist University, Kowloon Tong 999077, Hong Kong; 6CAS Key Laboratory of Health Informatics, Research Center of Biomedical Optics and Molecular Imaging, Institute of Biomedical and Health Engineering, Shenzhen Institutes of Advanced Technology, Chinese Academy of Sciences, Shenzhen 518055, China; yf.men@siat.ac.cn

**Keywords:** potassium sensor, *Arabidopsis*, root hair, tip growth, oscillating cytosolic K^+^

## Abstract

Root hair elongation relies on polarized cell expansion at the growing tip. As a major osmotically active ion, potassium is expected to be continuously assimilated to maintain cell turgor during hair tip growth. However, due to the lack of practicable detection methods, the dynamics and physiological role of K^+^ in hair growth are still unclear. In this report, we apply the small-molecule fluorescent K^+^ sensor NK3 in *Arabidopsis* root hairs for the first time. By employing NK3, oscillating cytoplasmic K^+^ dynamics can be resolved at the tip of growing root hairs, similar to the growth oscillation pattern. Cross-correlation analysis indicates that K^+^ oscillation leads the growth oscillations by approximately 1.5 s. Artificially increasing cytoplasmic K^+^ level showed no significant influence on hair growth rate, but led to the formation of swelling structures at the tip, an increase of cytosolic Ca^2+^ level and microfilament depolymerization, implying the involvement of antagonistic regulatory factors (e.g., Ca^2+^ signaling) in the causality between cytoplasmic K^+^ and hair growth. These results suggest that, in each round of oscillating root hair elongation, the oscillatory cell expansion accelerates on the heels of cytosolic K^+^ increment, and decelerates with the activation of antagonistic regulators, thus forming a negative feedback loop which ensures the normal growth of root hairs.

## 1. Introduction

Root hairs are tubular extensions from root epidermal cells, which facilitate water and nutrient uptake, as well as anchorage in the soil [1,2,3]. Root hairs develop in two main stages: initiation, when a small area of the epidermal cell wall loosens to form a swelling, and elongation, when the swelling grows in a polarized, turgor-driven fashion [2]. Once root hair elongation begins, exocytotic machinery, ion movement, and the cytoskeleton function together to maintain tip growth in a tube-like growth habit, leading to the elongated cylindrical morphology of mature root hairs [3,4].

A host of regulators, such as reactive oxygen species (ROS), calcium (Ca^2+^), and pH, have so far been shown to be involved in controlling the direction and growth rate of root hairs. Among these regulators, cytoplasmic Ca^2+^ gradients and oscillations have emerged as key components controlling the spatial dynamics of root hair elongation [5,6]. During root hair elongation, cytosolic Ca^2+^ levels are maximum (about 1 μM) at the apex and decline to a resting level (0.1–0.2 μM) at the base of root hairs [7,8]. Dissipating the tip-focused Ca^2+^ gradient has been shown to disrupt growth and manipulating the direction of Ca^2+^ gradient leads to redirected growth at the site where the new Ca^2+^ gradient is imposed [8,9], suggesting the direction of root hair elongation is controlled by the Ca^2+^ gradient. Furthermore, Monshausen et al. found that the cytosolic Ca^2+^ level oscillates with a similar frequency to growth rate, as well as ROS and pH changes, where the phase of Ca^2+^, ROS and pH oscillations lag behind the growth rate oscillations by several seconds [10,11]. These alternating oscillations are suggested to play vital roles in maintaining cell integrity and keeping the growth rate within tight boundaries during the rapid growth phase.

Potassium (K^+^) is another crucial ion which participates in root hair elongation. As one of the major osmotically active substances, K^+^ is expected to be continuously assimilated in hair cells, in order to maintain turgor during cell elongation. Disrupting the function of K^+^-selective ion channels by tetraethylammonium (TEA) has been shown to inhibit *Arabidopsis* root hair growth [12]. A complete loss-of-function mutant of TRH1 (AtKT3/AtKUP4), a member of the KT/KUP/HAK K^+^ transporter family, formed root hair initiation swellings without further tip elongation and exhibited a “tiny root hair” (*trh1*) phenotype [13,14]. AtAKT1, an inward rectifying potassium channel, has also been demonstrated to be required for hair growth and was implicated in potassium uptake during hair elongation [15,16]. Despite such findings demonstrating that K^+^ and K^+^ transporters/channels are specifically required for the elongation of root hairs, their biological functions and mechanisms remain to be defined.

In this study, we first apply the small-molecule fluorescent K^+^ sensor NK3 in *Arabidopsis* root hairs and find that the K^+^ level at root hair tip, as maintained by K^+^ transporters/channels, influences root hair growth. Further studies conclusively show that root hairs undergoing tip growth exhibit oscillating K^+^ levels, where the maximum of the K^+^ oscillation leads the growth oscillation by approximately 1.5 s. Treatments with high levels of extracellular K^+^ show no significant influence on hair growth, but trigger a swelling structure to form at the hair cell tip, induce cytoplasmic Ca^2+^ elevation, and disrupt actin cytoskeleton organization. Taken together, these findings suggest that cytosolic K^+^ oscillation may play a vital role in the feedback loop regulation of root hair elongation.

## 2. Results

### 2.1. Application of NK3 in Monitoring K^+^ Levels in Arabidopsis Root Hairs

To investigate cytoplasmic K^+^ dynamics in growing root hairs with high spatial and temporal resolution, a newly developed highly selective K^+^ fluorescent sensor, NK3, was used in this study (Appendix A) [17]. The optimal NK3 concentration was first determined by treating five-day-old green seedlings of Col-0 with different concentrations of NK3 and imaging under a Zeiss LSM 880 confocal microscope (Appendix A). The results showed that fluorescence intensity gradually increased with the incensement of NK3 concentration, approaching saturation at about 1 μM. Further toxicity experiments showed that 1 μM and 2 μM NK3 had no detectable effect on hair growth rates or final length (Appendix A), indicating that the analysis of root hair growth in NK3 stained plants was comparable with that of nontreated ones.

In order to verify whether NK3 was suitable for indicating the relative concentration of cytoplasmic K^+^ in root hair cells, NK3 was applied to root hairs of Col-0 grown on mediums supplemented with different concentrations of KCl and NaCl (Figure 1). The results showed that root hairs under long-term treatments of 25 mM and 50 mM KCl showed increased fluorescence intensity in root hair tips, whereas 25 mM and 50 mM NaCl decreased the fluorescence intensity, compared to ½ MS (Mock) treatments, consistent with previous reports that salinity lowers the K^+^ level in planta [18,19]. These observations demonstrate the utility of NK3 in measuring cytoplasmic K^+^ level in root hair cells, and that 1 μM is an optimal concentration for further experiments.

### 2.2. Relationship between Cytoplasmic K^+^ Level and Root Hair Growth

Numerous studies have reported that K^+^ transporters/channels are specifically required for K^+^ acquisition from soil [20,21]. Hence, different mutants of K^+^ transporters/channels were stained using NK3, in order to inspect whether NK3 is able to distinguish the differences in cytoplasmic K^+^ level among these mutants. We found that mutations of active inward K^+^ transporters/channels—AtKUP4 (TRH1) and AtAKT1—resulted in attenuated fluorescence intensities in root hair tips, while mutation of a silent K^+^ channel (AtKC1), a negative regulator of K^+^ uptake [22], exhibited enhanced fluorescence intensities (Figure 2A,B). These three K^+^ transporters/channels have all been identified in root epidermal cells, as well as hair cells, have been reported to participate in K^+^ acquisition [6]. In contrast, another mutation of K^+^ transporters—AtHAK5, which is only expressed in root epidermis and stele—showed no detectable influence on fluorescence intensity in root hairs compared to wild-type Col-0. Again, these results demonstrated that NK3 is applicable to monitoring the cytoplasmic K^+^ level in *Arabidopsis* root hairs.

Although, as an osmotically active substance, K^+^ is supposed to be assimilated in root hairs to drive cell elongation and as reports have shown that K^+^ transporters/channels participate in the regulation of root hair elongation [13,16], the relationship between hair growth and K^+^ or K^+^ transporters/channels remains unaccounted for, to date. By employing a seedling phenotyping platform (DYNAPLANT) [23], we examined the root hair elongation kinetics of different K^+^ transporters/channels mutants. Interestingly, similar to the variation of cytoplasmic K^+^ level, *kup4-2*, and *akt1* exhibited shorter hair lengths compared to that of Col-0 and *hak5* (Figure 2C and Appendix A), implying a positive correlation between root hair length and K^+^ level. Moreover, the time–course curvilinear hair growth rates revealed that the differences of final length were mainly dependent on varied growth rates at the mid-anaphase of hair elongation (Figure 2D). At the prophase of hair elongation, the mutant and wild-type root hairs all showed a gradual increase in growth rates. Then, after about 2.5 h, root hairs of *kup4-2* first reached their maximum rate (about 1.3 μm/min), while others continued to increase up to around 1.7 μm/min. These changes in growth rate were in line with the variation of cytoplasmic K^+^ level in the tested mutants. Taking all these results together, we conclude that root hair tip K^+^ homeostasis maintained by K^+^ transporters/channels controls the rapid growth of root hairs.

### 2.3. Cytosolic Level of K^+^ Oscillates during Root Hair Rapid Elongation

K^+^ transporters/channels likely function by adjusting K^+^ concentrations at the tip of the hair cell, in order to regulate cell turgor and influence root hair elongation. To further explore the relationship between cytoplasmic K^+^ level and root hair growth, we visualized K^+^ dynamics at the apex of growing root hairs based on supplementing the growth medium with 1 μM NK3. Along with the growth rates analysis, we were able to confirm that the apical K^+^ levels of growing root hairs of Col-0 oscillated at the same frequency (of two to four peaks per minute) as growth rates, whereas the K^+^ and growth oscillations were impaired at the apex of the *kup4-2* mutant root hairs (Figure 3 and Appendix A). While the magnitudes of both the oscillations in K^+^ and growth rate were variable, even within a single root hair, our measurements indicated a close temporal relationship between cytoplasmic K^+^ and tip growth, where each elevation of K^+^ level appeared to be followed by a rapid growth burst (Figure 3B). Cross-correlation analysis revealed that the oscillatory K^+^ levels most likely led tip growth oscillations by approximately 1.5 s (Figure 3C). When root hairs approached maturity, the growth rates were maintained at a very low level of 0.2 μm/min and cytosolic K^+^ levels gradually declined while weakening oscillations (Appendix A). Taken together, our results indicate that oscillations in cytosolic K^+^ levels precede root hair rapid elongation, which are coordinated by K^+^ transporters/channels.

### 2.4. Manipulating Cytosolic K^+^ Level Affects Root Hair Growth

To further validate the relationship between cytosolic K^+^ and tip growth, we attempted to manipulate cytoplasmic K^+^ levels while simultaneously tracking root hair cell tip elongation. Monitoring the fluorescence intensity of NK3 during exogenous K^+^ treatment showed that 100 mM KCl caused a rapid increase of fluorescence at the hair apex within a half-minute (Appendix A). The increase of cytoplasmic K^+^ may lead to an increase in cell osmotic turgor at the root hair tip and, thus, might promote cell elongation. However, we found that 100 mM KCl treatment did not show any influence on the growth dynamics of the hair cells within one minute (Appendix A), implying the existence of other factors limiting growth. Consistent with this hypothesis, we found that swelling structures formed at the root hair tips within 5 min among those root hairs treated with 100 mM KCl, which gradually restored to normal structure in the following 5–10 min, whereas 100 mM NaCl was not able to induce swelling formation at the root apex (Figure 4A,B). These observations indicate that while exogenous K^+^ treatment rapidly increased the level of cytoplasmic K^+^, cell structure and integrity regulators are probably also induced to restrict cell expansion, leading to the formation of swelling structures at root hair tips.

Cytoskeleton and Ca^2+^ have been reported to participate in the regulation of root hair tip growth [3,11]. Hence, we used *Arabidopsis* plants stably transformed with the GFP-based Ca^2+^ sensor Yellow Cameleon (YC) 3.6 [24] in order to monitor the changes of cytoplasmic Ca^2+^ level during the formation of the swelling structure after exogenous K^+^ treatment. We found that 100 mM KCl induced a significant increase of Ca^2+^ level at the root hair apex during the formation of swelling, whereas 100 mM NaCl induces a slight increase of Ca^2+^ without swelling formation (Figure 5, Appendix A). Furthermore, by employing microfilament marker lines (Lifeact–Venus), we found that the organization of actin microfilaments was also influenced by exogenous K^+^ treatment, but not Na^+^ treatment (Appendix A). Before treatment, the actin cytoskeleton was typically organized as thick actin bundles; however, during the formation of swelling structure, the organized actin cytoskeleton at the apex evanished. Then, the actin cytoskeleton reorganized during the recovery of the hair structure. Altogether, our results imply that K^+^ plays a more important role than osmotically active substances, and is closely interrelated with other crucial regulators of root hair elongation.

## 3. Discussion

### 3.1. Biological Significance of the Application of Fluorescent K^+^ Sensor in Arabidopsis

With a lack of practicable detection methods, the physiological role of K^+^ in plants—other than serving as one of the major osmotically active substances—still remains unclear [25,26]. The spatiotemporal patterns of physiological and pathological K^+^ changes in plant cells are yet to be unraveled. Compared to the traditional K^+^-selective microelectrode based-methods, fluorescent probe based-methods are expected to cause less damage to the cell, allowing the investigation of K^+^ dynamics with high spatial and temporal resolution [27,28].

Several synthetic fluorescent K^+^-sensitive sensors have been developed [27], but some of them suffer from Na^+^ influence [29,30], as well as low dissociation constant (*K*_d_) and dynamic range [31,32,33], which are not suitable for detecting K^+^ levels in plant cells [34,35]. In plant cells, physiological extracellular K^+^ concentrations are less than 10 mM, while intracellular K^+^ levels are usually around 100 mM [36]. Ideal intracellular K^+^ sensors that are applicable for cytoplasmic K^+^ level monitoring should have a relatively large *K*_d_ and dynamic range [37]. For this purpose, our group has developed and reported a series of K^+^-sensitive sensors, including NK3, with improved selectivity and dynamic range in recent years [37,38]; some KS probes have been applied in K^+^ dynamics monitoring in animal cells [37,39,40,41]. In this study, we first used *Arabidopsis* root hairs as a platform to verify the biological properties of the small-molecule fluorescent K^+^ sensor NK3 for cytoplasmic K^+^ level monitoring, and then applied NK3 to study the K^+^ dynamics in cell tip during root hair growth. By employing a variety of performance tests, we demonstrated the sound reliability of NK3 in root hair cells (Figure 1, Figure 2 and Appendix A) and laid a solid basis for various further investigations of K^+^ transport and signaling in plants, such as K^+^ transporter regulation, low-potassium tolerance, plant salt tolerance, and pollen tube polar growth.

### 3.2. K^+^ Dynamics and K^+^ Transporters/Channels May Play Vital Roles in Root Hair Tip Growth

Root hair tip growth is a complicated process which requires a high degree of coordination between cytoplasmic Ca^2+^ dynamics [5], apoplastic ROS and pH [42], the cytoskeleton [9], and vesicular trafficking [43]. Similar to tip growth oscillation, Ca^2+^, ROS, and pH have been shown to oscillate periodically, coupled with transient cell wall loosening [44,45,46]. The maxima of the oscillatory fluctuations in Ca^2+^ lag behind hair cell growth rate peaks by approximately 5 to 6 s, while ROS and pH oscillations lag behind growth oscillations by 7 to 8 s [10,11]. Even though studies have suggested that Ca^2+^, ROS, and pH regulation of growth form a positive feedback loop to sustain the tip growth of root hairs [3,47], it remains unclear whether there are other factors involved in the polar growth of root hairs.

In this study, we confirmed that K^+^ transporters and channels are involved in root hair elongation (Figure 2), as well as demonstrating that TRH1, a member of the KT/KUP/HAK K^+^ transporter family, plays a critical role in supporting hair cell tip growth. We also monitored cytosolic K^+^ dynamics during hair cell tip growth, and found that K^+^ oscillates with a similar frequency to the growth rate, but precedes growth oscillations by approximately 1.5 s (Figure 3). Furthermore, we found that the level of cytosolic Ca^2+^ at the root hair apex can be induced by exogenous KCl treatment (Figure 5 and Appendix A). Therefore, it is possible that cytoplasmic K^+^ oscillations, along with intracellular Ca^2+^ and extracellular ROS and pH changes, may be linked in a feedback system that regulates tip growth after root hair cell initiation. During each growth pulse, hair cells assimilate soil K^+^ through activated K^+^ transporters, leading to the acceleration of growth, followed by activated cyclic nucleotide-gated channels (CNGCs) elevating the cytosolic Ca^2+^ level which, subsequently, trigger apoplastic ROS and pH response systems to slow down tip growth, and activate K^+^ transporters to support the next pulse of growth oscillations (Figure 6). Such a model fits well with the idea that some calcineurin B-like protein (CBL) family members which interact with CBL-interacting protein kinases (CIPKs) function to regulate the activity of K^+^ transporters/channels and responses to Ca^2+^ dynamics and K^+^ deficiencies in plants [48,49]. Therefore, the identification of K^+^ transporter/channels involved in the regulation of K^+^ oscillation, as well as their relationships with Ca^2+^ signaling in growth oscillation, are major challenges for future research [50,51].

## 4. Materials and Methods

### 4.1. Plant Material and Growth Conditions

All plant material employed in this study were *Arabidopsis* Col-0 wild-type plants or mutants in the Col-0 background. Transgenic marker lines and K^+^ transporters mutants were as follows: YC3.6 [24], Lifeact–Venus [52], *kup4-2* (SALK_071644) [53], *akt1* (SALK_071803) [49], *atkc1* (SALK_140579) [54], and *hak5* (SALK_005604) [55], which have been described previously. Surface-sterilized seeds were sown on half-strength Murashige and Skoog (½ MS) medium (2.2 g/L MS salts, 15 g/L sucrose, pH 5.7–5.8, and 1 g/L phytagel) and germinated at 22 °C with a 16 L/8 D illumination cycle after imbibition. Five-day-old green seedlings were chosen for experiments.

### 4.2. Sensor Solution

NK3 is a newly developed small-molecule fluorescent K^+^ sensor (Appendix A) [17]. The 2 mM stock solutions of NK3 were prepared in DMSO.

### 4.3. Confocal Microscopy

Five-day-old green seedlings were transferred to purpose-built cuvettes and mounted as described previously [10]. After 12 h of growth in ½ MS medium, root hairs were imaged with the Zeiss LSM 880 laser scanning confocal microscope (Carl Zeiss, Oberkochen, Germany) using a 40× water immersion, 1.2 numerical aperture, C-Apochromat objective. NK3 was excited using a 561 nm laser, and emission was collected from 580 nm to 650 nm. Bright-field images were acquired simultaneously using the transmission detector of the microscope. For time-lapse analysis, images were collected every 3 s. The Ca^2+^ imaging acquisition protocol was described as follows. The YC3.6 Ca^2+^ sensor was excited at 458 nm. The CFP emission was collected at wavelengths from 465 to 505 nm, while FRET emission was collected at wavelengths from 520 to 540 nm. For the observation of actin filaments visualized by Lifeact–Venus in root hair, samples were observed under 40× water immersion. Serial confocal images were obtained every 0.5 μm. The fluorescence intensity was measured using the Image J software (Wayne Rasband, NIH, Bethesda, MD, USA).

### 4.4. Measurement of Root Hair Growth

To observe the effect of KCl and NaCl on root hair growth, growth measurements were performed on root hairs immersed in liquid half MS medium. After observing the growth of a root hair for several minutes before treatment, the reagent (100 mM KCl or NaCl) was gently mixed into the medium and growth measurements were continued on the same cell. The use of liquid medium allowed noticeable shifting of the root axis and made measurements of root hair tip growth more difficult.

Cross-correlation analysis was performed to determine the temporal relationship between K^+^ oscillations and growth oscillations. The correlation coefficient ($r=\;\frac{SP_{xy}}{\sqrt[{2}]{SS_{x}SS_{y}}}$) was determined as the measurements of the growth oscillations were shifted in time with respect to the growth oscillations. *SS_x_* and *SS_y_* are the sum of squares for corresponding K^+^ and growth recordings, while *SP_xy_* is the sum of the products of the two corresponding recordings. The temporal resolution of the analysis was the same as that used to acquire the images, but with an offset of one-half of the temporal resolution due to the fact that the growth rate measurements were plotted at the halfway point between the corresponding images used to determine the growth rate.

### 4.5. Root Elongation Kinetic Assays

Root elongation kinetic assays were carried out by a commercial seedling phenotyping platform (DYNAPLANT, Microlens, Beijing, China) with high throughput (up to 300 roots per 5 min) and a spatial resolution of 1.2 μm per pixel (http://www.yph-bio.com/DynaPlant.asp) [23]. The 4.5-day-old seedlings were transferred carefully onto new plates of growth medium with respective chemicals 12 h prior to analysis, in order to saturate incubation. Plates were docked onto the platform and subjected to 5 min interval imaging. All images were automatically analyzed using the DYNAPLANT software.

## Figures and Tables

**Figure 1 ijms-21-06184-f001:**
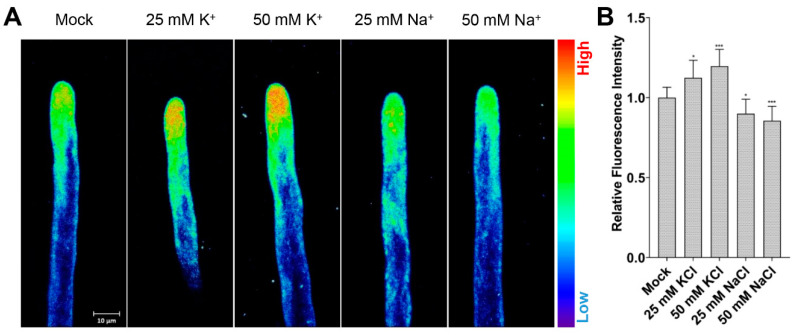
NK3 fluorescence intensity varies with different K^+^ concentrations in *Arabidopsis* root tips: (**A**) NK3 fluorescence intensity images of 5-day-old Col-0 root hairs immersed in ½ MS medium containing 1 μM NK3 (Mock) or 1 μM NK3 supplemented with different concentration of KCl or NaCl (25, 50 mM) for 12 h and visualized using pseudocolor; (**B**) Quantification of the fluorescence intensity of 5-day-old Col-0 root hairs immersed in ½ MS medium containing 1 μM NK3 (Mock) or 1 μM NK3 supplemented with different concentrations of KCl or NaCl (25, 50 mM) for 12 h. Bars represent the average intensity (mean ± s.d.) of fourteen hairs. NK3 fluorescence was measured in approximately 30 μm^2^ regions at the apex of root hairs (Student’s *t*-test between Mock and treated seedlings; * *p* < 0.05, *** *p* < 0.001).

**Figure 2 ijms-21-06184-f002:**
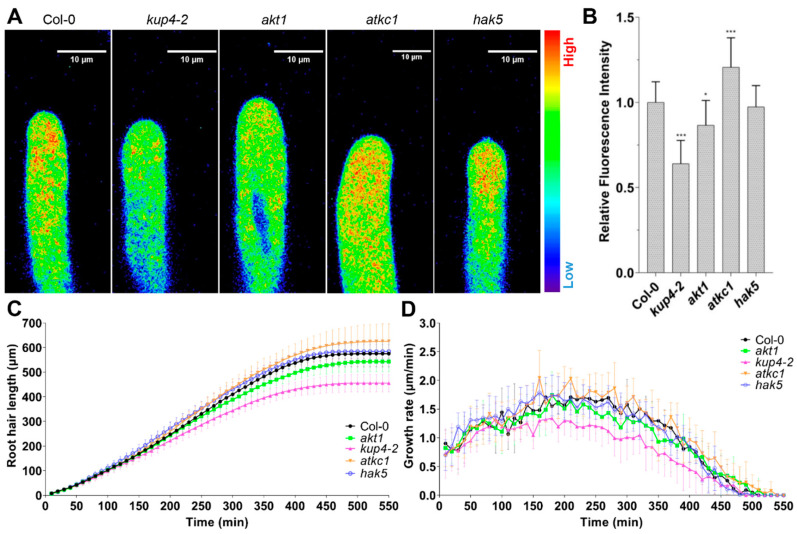
Mutants of K^+^ transporters/channels show altered NK3 fluorescence intensity and root hair growth: (**A**) NK3 fluorescence intensity images of root hairs of Col-0, *kup4-2*, *akt1*, *atkc1*, and *hak5* immersed in ½ MS medium containing 1 μM NK3 and visualized using pseudocolor; (**B**) Quantification of the fluorescence intensity in A. Bars represent the average intensity (mean ± s.d.) of fourteen hairs (Student’s *t*-test between Col-0 and mutant seedlings; * *p* < 0.05, *** *p* < 0.001); (**C**) Growth curve of root hairs of 5-day-old green seedling of Col-0, *kup4-2*, *akt1*, *atkc1*, and *hak5* transferred on ½ MS medium. Bars represent the average length (mean ± s.d.) of twelve hairs; (**D**) Growth rate curve of root hairs in C.

**Figure 3 ijms-21-06184-f003:**
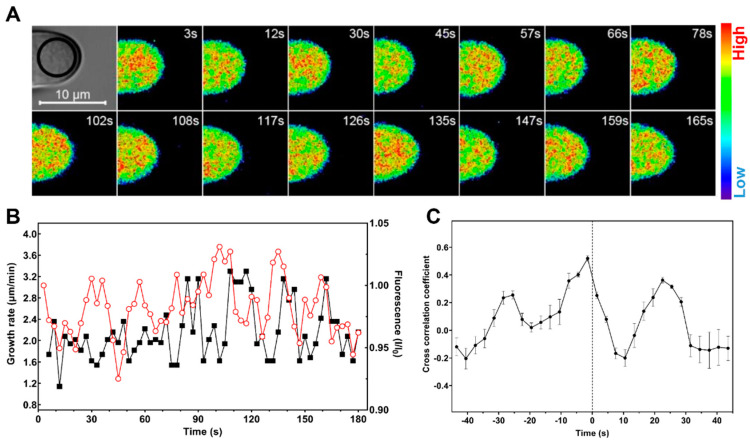
*Arabidopsis* root hair tips show oscillating K^+^, which peaks ahead of maximal growth: (**A**) Time-lapse NK3 fluorescence intensity images of Col-0 root hairs undergoing tip growth and visualized using pseudocolor; (**B**) Quantitative analysis of root hair growth rates (filled squares) and NK3 relative fluorescence (I/I_0_; open circles) at the root hair apex. NK3 fluorescence was measured in approximately 30 μm^2^ regions indicated in (**A**). Representative results of eight measurements are shown; (**C**) Cross-correlation analysis of fluorescence oscillations with growth oscillations indicates that the increases in growth rate lag behind the increases in cytosolic K^+^ by approximately 1.5 s. Cross-correlation was performed on data from four separate root hairs.

**Figure 4 ijms-21-06184-f004:**
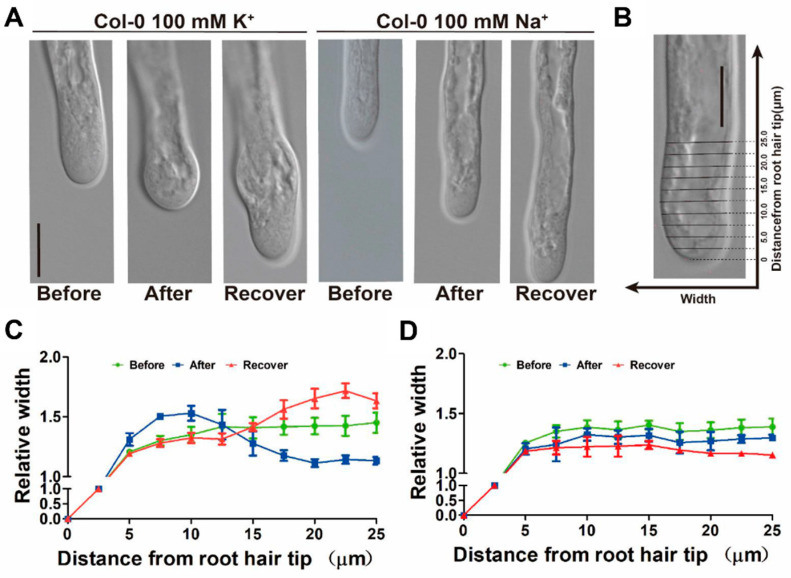
Effects of exogenous K^+^ on the growth of *Arabidopsis* root hairs: (**A**) Time-lapse bright-field images of root hairs under 100 mM KCl or NaCl treatment. After K^+^ treatment, a swelling structure formed at the root hair tips within 5 min and recovered to normal structure in the following 5–10 min. Under the same concentration of Na^+^, there was no such structure. Bar = 10 μm; (**B**) The representative scheme of measurement of root hair width at a certain position from the root hair tip. Bar = 10 μm; (**C**,**D**) Quantification of the width of root hairs in A. Curvature represents the average width at different positions (mean ± s.d.) in six hairs. (**C**) Under 100 mM KCl treatment and (**D**) under 100 mM NaCl treatment.

**Figure 5 ijms-21-06184-f005:**
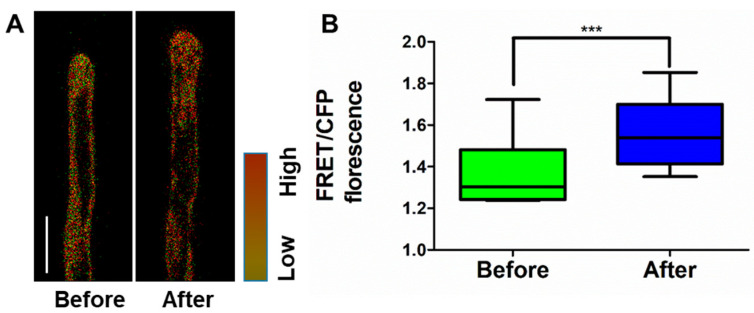
K^+^ application influences the cytoplasmic Ca^2+^ level at root hair tips: (**A**) 100 mM KCl treatment induces the increase of cytoplasmic Ca^2+^ level in root hairs. Root hairs from wild-type expressing the Ca^2+^ sensor YC3.6 were imaged. The red channel represents the intensity of FRET and the green channel represents the intensity of CFP. Bar = 20 μm. Before, root hairs before treatment. After, swelling structures formed at the root hair tips within 5 min after treatment; (**B**) Quantitative analysis of cytosolic Ca^2+^ growing root hair from the wild-type. Bars represent the average length (mean ± s.d.) of six hairs (Student’s *t*-test, between seedlings before and after treated; *** *p* < 0.001).

**Figure 6 ijms-21-06184-f006:**
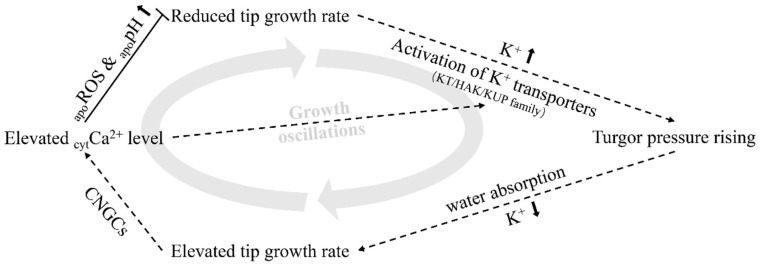
Working model: Activated K^+^ transporters promote the assimilation of soil K^+^, elevating the cell turgor pressure and, leading to water absorption and growth acceleration. Then, the root hairs acquire Ca^2+^ through cyclic nucleotide-gated channels (CNGCs), where the increase in Ca^2+^ level triggers apoplastic ROS (_apo_ROS) production, alkalizes apoplastic pH (_apo_pH), and restricts tip growth; likely also activating K^+^ transporters to prepare for the next round of growth oscillations.

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
