# Peer review of "Close Temporal Relationship between Oscillating Cytosolic K+ and Growth in Root Hairs of Arabidopsis"

_ijms, 2020, doi:10.3390/ijms21176184_

Round 1

Reviewer 1 Report

General comments:

This manuscript described the implementation of a new K+ sensitive dye to quantify cytoplasmic K+ dynamics in growing Arabidopsis root hairs. In previous articles, the authors have displayed their efforts in generating novel K+ sensors. More so, they have previous shown the applicability of those sensors in vivo. To my knowledge, this is their first report showing the usability of the respective NK3 sensor in planta. Whereas the involvement of K+ signaling during plant cell growth has been illustrated before, the tools to further decipher fine K+ dynamics during growth have been missing. Hence, the work presented here is relevant to the plant science community.

The manuscript is relatively well written and to the point, but could use a revision by a native English speaker. Based on a thorough set of experiments, the authors have shown that NK3 might be a useful tool to decipher K+ dynamics at high spatio-temporal resolution. I strongly appreciate the effort put into the growth rate analysis and the incorporation of several K+ transport mutants.
Importantly however, the outcome and/or presentation of some experiments does spark some doubt about the degree of responsiveness of NK3 to [K+]cyt in planta. Whereas the in vitro NK3 calibration clearly shows strong K+-dependent fluorescence changes, the extent to which NK3 fluorescence is affected in root hairs (even upon 100mM KCl supplementation) seems rather low. More specifically, whereas Fig. 1A and 2A,B show reasonable K-dependent signal increase, Fig. 3A,B, Fig. S6 and the suppl. movie suggest otherwise. This is especially important because, as a result, substantial proof for the existence of consistent cytosolic K+ oscillations remains somehow doubtful. To make the data objectively interpretable for the reader, I’ve made some suggestions to the authors to display and/or analyse their data in an alternative manner.
Similarly, the data used to propose a link between [K+]cyt, [Ca2+]cyt and actin dynamics remains subjective, needs reworking and additional analysis. In its current state, there is no empirical evidence convincingly showing this link. Given the potential implications of having a working K+ sensor with the Kd and dynamic range needed to visualise cytosolic K+ dynamics in planta, I would be happy to see how the data looks after thorough revision.

Finally, the authors repeatedly postulate that K+ dynamics could regulate turgor pressure changes during root hair growth. Whereas this might be true, there is no direct evidence supporting this claim. Therefore, and in some other cases described hereafter, I would suggest to tone down some statements made in the text.

Detailed comments:

line 23: ‘supposed to be’ is suggestive. ‘expected to be’ is more mild.

Line 27: remove ‘an’

Line 31-32: seems like a premature conclusion based on the data that has been presented here. There are numerous ways by which K+ elevation could affect cell elongation and cytoplasmic calcium dynamics. The dynamic nature of cytoplasmic calcium transients and oscillations and its degree of causality with respect to growth rate dynamics remains poorly understood.

Line 32-35: These statements feel premature. The data presented here is descriptive and does not directly prove nor suggest that K+ dynamics relate to turgor during root hair growth. Toning down the speculative turgor-related statements throughout the manuscript would improve overall credibility.

‘before each burst’. Unclear what type of burst the authors are talking about.

I also feel that it’s premature to state that K+ oscillations are ‘restricted by antagonistic regulators’. The data to support this is very limited.

Line 56: ‘restricting’ makes it seem as if these oscillations are there to keep a brake on growth rate only, whereas the oscillatory machinery in fact both ‘promotes’ growth and ‘restricts’ growth. It would be more correct to state e.g. ‘keep growth rate within tight boundaries’.

Line 58: ‘ion function’ please rephrase

Line 59: ‘supposed to be’ see previous comment

Line 72-74: Without prior knowledge, it’s already very surprising to read that K+ elevation would affect root hair morphology, [Ca2+]cyt and actin, while not affecting growth. I believe the authors should carefully revise their data on this, and assess whether their spatial resolution, tip tracking protocol, calcium and actin analysis justify making such conclusions.

Line 79/Fig. S1: The authors claim K+ specificity based on the inability of NK3 to respond to Na+ and a set of metal ions. Importantly, the authors should at least also show pH and ROS responsiveness.

Fig. S3B: X-axis label states ‘KS23’ concentration

Fig. 1: the relative fluorescence increase at 25 and 50 mM KCl treatment is surprisingly small compared to the signal intensity observed under control conditions and compared to the in vitro calibration results shown in supplementary. Can the authors comment on why this might be?

Line 106: ‘researches’ please replace with correct term

Line 108: ‘testify’ please rephrase

Fig. 2C: could the authors please include supplementary images of the root containing the root hair growth and mature zone for each of the mutants?

Line 136-141: I appreciate the authors effort to provide a detailed analysis of the growth rate kinetics. Importantly however, the interpretation of Fig. 2D is subjective and can easily be improved by calculating a fit for each individual root hair time series, and objectively extracting the ‘tipping point’ (the timepoint at which the growth rate starts decreasing again) from each fit using the equation. This data could then be analysed statistically.

Line 141-142: Since no correlation analysis of growth rate and [K+] in time was performed, I would suggest to not use the term ‘positive correlations’. Instead, it would be more correct to state that the findings on growth rate are in line with the overall K+ levels.

Movie S1/Fig. 3A-B: the authors show a pseudocolored timelapse acquisition of K+ dynamics in a growing WT root hair and the analysis thereof. Whereas there is some noticeable variation in signal intensity throughout the acquisition, it’s very difficult to judge whether or not they relate to clear oscillations in K+ level. Hence, the movie and Fig. 1A don’t convincingly show K+ oscillations. I would like to make the following suggestions:

  • replace Fig. 1A by a kymograph, which should allow the reader to easily interprete an entire time-series at first sight.
  • Show kymographs for WT and kup4-2 side by side. The lack of K+ oscillations needs to be shown in the main manuscript
  • Movie S1 should show a growing WT and kup4-2 root hair side by side.
  • Adjust the pseudocolor parameters in such a way that one can more easily distinguish a K+ trough compared to a K+ peak.
  • For Fig. 1B, please include the same analysis for kup4-2

Fig. S4: it would be very interesting to see whether or not kup4-2 root hair still display growth rate oscillations.

Line 176: ‘thus probably’ replace by ‘might thus’

Line 173-174: The relative fluorescence increase upon treatment with 100 mM KCl is very low, especially compared to Fig. 1 and compared to what one would expect based on the NK3 calibration performed earlier (strong K+ dependent fluorescence increase). Can the authors comment on why that would be?

Line 176-177: It is very surprising to see that KCl treatment results in root hair bulging, but doesn’t seem to affect root hair growth rate. Transient bulging generally coincides with a transient decrease in growth rates. Can the authors comment on this? Is this a discrepancy which might be due to the difficult imaging conditions upon treatment in liquid medium? If so, I would suggest to remove the growth rate analysis in this case. Precisely monitoring root hair growth rate changes requires registration algorithms with subpixel accuracy which can’t be achieved when there is xyz shift of the sample during acquisition.  

Line 178-184: the authors attribute root hair swelling to a K+-induced increase in turgor pressure and the activation of the cell integrity machinery to prevent bursting. While theoretically plausible, I feel that the data presented here is too preliminary to allow such interpretations or suggestions. As a reader, I would feel more comfortable if the authors would simply present their data in a descriptive manner, without jumping to conclusions on why these changes might occur. Hence, please rephrase or delete line 181-184.

Line 195: The statement that a calcium increase is expected to limit growth should be rephrased. Suggesting a linear relationship between calcium increase and growth arrest somehow oversimplifies the complexity of the calcium signaling pathway.

Line 198-199: I’m not convinced by this analysis. Fig. 5A does not show a clear [Ca2+]cyt increase. In fact, it seems to show the opposite with a higher calcium level in the pre-treatment root hair. Importantly, root hair bulging (as depicted here) generally coincides with a decrease in apical calcium levels (as Fig. 5A suggests).
Also, Fig. S8 shows a pre-treatment timeframe of only 100s. In spite of being a ratiometric FRET sensor, the FRET ratio still tends to vary during acquisition (e.g. bleaching-like behaviour or increase) due to e.g. differences in CFP and YFP quantum yield etc. This is especially true when the signal to noise ratio for one or both channels is too low. That being said, in this particular case, I would suggest to authors to include a longer pre-treatment phase (which would allow better assessment of the native FRET variation during acquisition) and independently show the CFP, YFP and ratio traces. By doing so, the reader gets all the information that is needed to objectively judge the outcome of these experiments. In its current form however, I tend to be conservative on the involvement of [Ca2+]cyt.

Line 199: ‘maker’ correct to ‘marker’

Line 199-203/fig. S9: a more pragmatic analysis approach would be needed here to clearly show whether or not the actin cytoskeleton dynamics are truly affected. In this case it might again be a good idea to construct kymographs for each acquisition. This could provide an illustrative and quantifiable basis for assessing F- and G-actin position relative to the root hair tip.

Line 251-255: This is indeed an appealing hypothesis. I’m curious to see the outcome of the calcium analysis once it has been revised.

Discussion: Can the authors make some suggestions on how they would study the causality of root hair turgor pressure, K+ and growth rate oscillations? Can they suggest a way to quantify turgor pressure ‘changes’ during growth?
Can the authors also provide some insight into what is known (or not known) about turgor pressure and the dynamics thereof during oscillatory cell growth?

The presence of an apical cytosolic K+ gradient has previously been demonstrated by Halperin and Lynch (2003) using PBFI. Can the authors comment on why they choose not to further explore that specific dye to resolve temporal dynamics?

Line 295: It would be nice to get some more detail on how the authors extracted root hair growth rate oscillations from their acquisitions. Overall there’s little information about how the acquisitions (K+, Ca2+, growth rate) were analysed. E.g. how was the apical ROI defined, and how did the authors measure the same ROI from frame to frame. Did they perform a slice registration/template mating for example?

Line 307-309: Strongly appreciated that the authors kept that in mind. This is crucial, especially when it comes to quantifying phase shifts.

Reviewer 2 Report

Comments to the Authors

In the study described in this manuscript, the authors employed a small-molecule fluorescent K+ sensor, NK3, to determine the dynamics of cytoplasmic K+ in Arabidopsis root hairs for the first time. They suggest that in the regulation of root hair elongation, the oscillatory cytosolic K+ levels provide the turgor driving the expansion of cells before each burst, and that this is restricted by antagonistic regulators to form a negative feedback loop to ensure the normal growth of root hairs. This manuscript contributes significantly to our knowledge of the mechanisms regulating the growth of root hairs. However, I notice several issues in the manuscript, which I have listed in the following comments for the authors’ consideration:

Major comments:

  1. Figure 1B, 2B, S3B: Why is the fluorescence intensity (y-axis) significantly different between the graphs in Figure 1B and Figure 2B or Figure S3B under the same conditions?
  2. Figure 2: The authors have shown a relationship between the uptake of K+ and elongation of root hair using K+transporter/channel mutants. I wonder what happens to the expression of KUP4, AKT1, ATKC1, and HAK5. Is their expression increased during the elongation of root hairs upon K+ uptake?
  3. L29–30, 73–74, 198–199; Figure 5: If the authors contend that K+ induces the uptake of Ca2+, then a negative control experiment, using Na+ or any other osmotically active substance, as shown in Figure 4, should be performed.
  4. L29–31, 74; Figure S9: As I have mentioned in comment 3, if the authors contend that K+ levels lead to the depolymerization of microfilaments, then a negative control experiment, using Na+ or any other osmotically active substance, as shown in Figure 4, should be performed. Alternatively, a control experiment with K+ transporter/channel mutants (kup4, akt1, atkc1, and hak5) should be performed.

Minor comments:

  1. L82, L85: The notations of Figures S2 and S3 in the text do not match with the data actually presented in these figures.
  2. The legend of Figure S3 is incorrect. In addition, the title on the x-axis of the graph presented in Figure S3B is incorrect. The authors should cross-check the legends of the figures.

Round 2

Reviewer 1 Report

In the manuscript "Close Temporal Relationship between Oscillating Cytosolic K+ and Growth in Root Hairs of Arabidopsis" Sun et al. demonstrated the use of a new K+ sensor to monitor cytoplasmic K+ in Arabidopsis root hairs. This study brings novel insights into K+ dynamics during polar growth and is of considerable interest to the scientific community.

The authors answered the questions satisfactorily and made sufficient changes to the manuscript. Hence, I do recommend this paper for publication in its current form.

Reviewer 2 Report

I believe the manuscript has been significantly improved.